# Unraveling the Heterogeneity of ALS—A Call to Redefine Patient Stratification for Better Outcomes in Clinical Trials

**DOI:** 10.3390/cells13050452

**Published:** 2024-03-05

**Authors:** Laura Tzeplaeff, Alexandra V. Jürs, Camilla Wohnrade, Antonia F. Demleitner

**Affiliations:** 1Department of Neurology, Rechts der Isar Hospital, Technical University of Munich, 81675 München, Germany; 2Translational Neurodegeneration Section “Albrecht Kossel”, Department of Neurology, University Medical Center Rostock, 18057 Rostock, Germany; 3Department of Neurology, Hannover Medical School, 30625 Hannover, Germany; wohnrade.camilla@mh-hannover.de

**Keywords:** amyotrophic lateral sclerosis, clinical trials, heterogeneity, subcluster

## Abstract

Despite tremendous efforts in basic research and a growing number of clinical trials aiming to find effective treatments, amyotrophic lateral sclerosis (ALS) remains an incurable disease. One possible reason for the lack of effective causative treatment options is that ALS may not be a single disease entity but rather may represent a clinical syndrome, with diverse genetic and molecular causes, histopathological alterations, and subsequent clinical presentations contributing to its complexity and variability among individuals. Defining a way to subcluster ALS patients is becoming a central endeavor in the field. Identifying specific clusters and applying them in clinical trials could enable the development of more effective treatments. This review aims to summarize the available data on heterogeneity in ALS with regard to various aspects, e.g., clinical, genetic, and molecular.

## 1. Introduction

Amyotrophic lateral sclerosis (ALS) is a progressive neurodegenerative disorder characterized by the degeneration of motor neurons in the central and peripheral nervous system, leading to muscle weakness, atrophy, and eventually complete paralysis. Despite a high number of potential drugs being successful at the preclinical level, most clinical trials aiming at causative or symptomatic treatment have failed to demonstrate significant efficacy [1].

One major insight might be the concept that ALS is not a unique disease entity but rather a clinical syndrome, with diverse genetic causes, clinical presentations, and histopathological and molecular alterations contributing to its complexity and variability among individuals [2,3,4]. Thus, this heterogeneity poses challenges in understanding and treating ALS comprehensively and raises important questions: If patients cannot be treated uniformly as a whole but instead need individualized therapeutic approaches, how can subclusters that respond to certain approaches be identified in clinical trials? If and to what extent can these subclusters aid in improving the precision and effectiveness of therapeutic strategies?

In this aim, this review will discuss the current knowledge on heterogeneity in ALS with particular focus on clinical, genetic, pathological, and molecular aspects and outline how heterogeneity could be considered in basic research, clinical trials, and personalized medical approaches. An overview of the different aspects of heterogeneity in ALS discussed in this review is represented in Figure 1 and summarized at the end of the review in Table 1.

## 2. Genetic and Related Pathophysiological Heterogeneity

### 2.1. Genetic Architecture of ALS

One of the first striking pieces of evidence of ALS heterogeneity is its complex genetic architecture (Figure 1a). Research in the field of ALS-linked genes is progressing rapidly due to genome-wide association studies and next-generation sequencing techniques. Approximately 10% of ALS patients are classified as familial ALS (fALS), since they have at least one other affected family member with a usually autosomal-dominant pattern of inheritance. Ninety percent of ALS cases are classified as sporadic ALS (sALS). Both fALS and sALS can be linked to genetic mutations. A review from 2019 reported that 55.5% of fALS cases are caused by variants in known ALS-linked genes in the European population [5]. Herein, *C9orf72*-associated cases represent the most prominent fALS-associated gene alteration with 33.7%, followed by *SOD1* with 14.8%, *TARDBP* with 4.2%, and *FUS* with 2.8%. Although less frequent, in European cases of sALS, a repartition of 5.1% *C9orf72* cases, 1.2% *SOD1* cases, 0.8% *TARDBP* cases, and 0.3% *FUS* cases is observed (Figure 1a). When looking at the Asian population, over 30% of the fALS cases are associated with variants in the *SOD1* gene, highlighting notable ethnical differences in the genetic background of ALS. In 44.5% of fALS and 92.6% of sALS, no mutation of the four main ALS linked genes (*C9orf72*, *TARDBP*, *SOD1*, or *FUS*) are found [5]. Adding to the complex genetic architecture of ALS, we currently count approximately 50 genes that are known to be linked to ALS and a further ~100 genes that have been proposed to contribute as risk factors or act as disease modifier genes [4,5]. Different mutations in a single causative gene have further been associated with different and overlapping phenotypes regarding symptoms or disease progression and survival, further complicating the overall picture [6,47,48]. To date, the list of gene variants characterized as being pathogenic (class 5) or likely pathogenic (class 4), as described by the American College of Medical Genetics and Genomics (ACMG) guideline, has been constantly updated [49]. Interestingly, twin studies estimated the heritability of sALS being approximately 60%, highlighting the role of genetics in the course of the disease [50]. The genetic complexity of ALS could be explained at least in parts by an incomplete penetrance of ALS-associated genes as well as by the combination of genetic variants and environmental risk factors [51,52].

### 2.2. Genetic Subtypes Are Associated with Specific Pathophysiology

ALS is associated with a complex array of dysregulated interconnected processes, including glutamate excitotoxicity, immune system dysfunction, axonal transport defects, mitochondrial dysfunction, aberrant RNA processing, and many other pathways [4]. First insights into the different involved pathomechanisms were gained upon discovery of the first causative mutations. Considering different mutations, some seem to be more involved in specific cellular pathomechanisms than others, and we will only detail a few examples in this section. As the first example, most of the *FUS* mutations are in the nuclear localization sequence and therefore typically involved in dysfunctional nucleocytoplasmic shuttling. While fALS cases with a *FUS* mutation represent an aggressive disease with early onset, the nuclear import dysfunction in vitro is evident in early disease stages and correlates inversely with the age of onset [53,54]. In patients with a *C9orf72* mutation, disease onset is typically later than in *FUS* ALS [51], and *C9orf72*-mutated iPSCs accordingly present the dysregulated proteins needed for the nucleocytoplasmic transport, particularly in later stages. This might either be traced back to a later involvement of protein transport dysfunction in *C9orf72* ALS, or display a general age-dependent effect, even though the link to the age of onset in *C9orf72* patients has not yet been investigated [55]. As a second example, *C9orf72* and *SOD1* variants both feature high levels of oxidative stress, and as a result, CarboxyEthylPyrole (CEP) accumulates in the brain [56]. It is a fact that oxidative stress level is increased in brain tissue of *SOD1* ALS patients, but this could also be attributable to other factors than *SOD1* activity itself. Indeed, *SOD1* ALS patients particularly exhibit high CEP levels in brain samples with >90% CEP-positive neurons and ~5-fold higher levels of CEP deposition compared to those from *C9orf72* ALS and ALS without a known mutation. This is attributed to the amount of myeloperoxidase (MPO), as substantially more myeloperoxidase-expressing cells were detected in *SOD1* (~8-fold) and *C9orf72* (~2-fold) ALS patients’ brains compared to controls [56]. It must be emphasized that the SOD1 activity is independent of this phenomenon as it does not correlate clinically with the aggressiveness of ALS [57]. As a last example, Theunissen and colleagues identified a polymorphism in the ALS-related *STMN2* gene (11.5% of sALS in this study) in a cross-sectional case-control analysis [58,59]. Stathmin-2 protein is abundant in motor neurons and plays an important role in microtubule dynamics, axonal transport, stability, and regeneration, as well as for neuromuscular junction stability [60]. The reported case-control study supports the hypothesis that allele length is a determinant of disease risk and that stratification into risk genotype groups might be a useful tool for cohort selection.

Their data support the hypothesis that allele length is a determinant of disease risk and that stratification into risk genotype groups might be a useful tool for cohort selection. Above all, the group with two long cytosine–adenine repeat alleles (L/L 24 CA) was associated with a reduced survival and a severe functional decline.

### 2.3. Treatment Options Targeting Different Genetic Subtypes

Treating ALS patients according to their genetic status is becoming more common. The targeted antisense oligonucleotide (ASO), Tofersen, reduces the expression of pathogenic SOD1 protein, which represents a hallmark of pathology in cases associated with a gain in function mutation in the *SOD1* gene [61]. Differences in treatment response vary within a subgroup of mutation carriers due to large heterogeneities. For example, some ALS patients with *SOD1* mutations exhibit no toxic gain of function pathogenesis and probably do not benefit from the reduced SOD1 levels in Tofersen treatment. Similar treatment strategies are currently under development for ALS patients with *FUS* and *ATXN2 mutations* (NCT04768972, NCT04494256) and might further revolutionize the ALS treatment landscape. Another ASO therapy concerns Stathmin-2 (*STMN2*) (NCT05633459), which acts as an important splice target of TDP-43 (TAR DNA-binding protein 43), and thus the loss of nuclear TDP-43, as observed in ALS leading to a Stathmin-2 mis-splicing. While the clinical trial of this ASO therapy is ongoing, the Stathmin-2 levels have been restored in preclinical studies and could transform the therapeutical options for ALS patients [62,63]. The failure of the C9orf72 trials (NCT03626012, NCT04931862) revealed the indispensability of understanding the precise pathophysiological mechanism of distinct subgroups, although the exact reason for this failure is still unknown [62]. Hence, new therapeutic approaches targeting particular mechanisms involved in specific mutations are important in addition to ASO treatments. One of these approaches is the treatment with Arimoclomol (NCT00706147) in ALS patients with a *SOD1* mutation leading to an unstable SOD1 protein. Arimoclomol promotes the refolding of misfolded SOD1, but only a limited clinical effect was seen in the small patient cohort [64]. In two clinical trials of Verdiperstat (NCT 04297682, NCT04436510), an inhibitor of MPO targeting CEP production (previously discussed in the Section 2.2), no clinical benefit was observed in the general ALS cohort. However, no subclassification or genetic testing for *SOD1* mutation carriers, in whom CEP production plays a central role, was performed. Another study targeted neurotransmission by addressing the risk variant *UNC13A*. While clinical trials found no difference in the overall ALS cohort after lithium treatment, post-hoc analyses revealed extended survival in ALS patients carrying a specific homozygous polymorphism in the *UNC13A* gene [65]. A randomized, event-driven, double-blind, placebo-controlled trial studying the effect of lithium in ALS patients genotyped for the *UNC13A* polymorphism is ongoing [66].

## 3. Pathological Heterogeneity

Postmortem examination of brains and spinal cords from ALS patients reveals diverse patterns of motor neuron loss and associated pathological features (Figure 1b). Almost 97% of all sALS and fALS cases display inclusion bodies with cytoplasmic hyperphosphorylated TDP-43, marked for ubiquitination, being the major component of those inclusions [8,67]. However, *C9orf72* ALS postmortem tissue shows an accumulation of p62-positive aggregates caused by abnormal dipeptide repeat (DPRs) proteins besides TDP-43 inclusions [7]. Based on the composition, morphology, and presence of TDP-43 pathology, Tan et al. [9] described three ALS-TDP-43 subtypes in the motor cortex. Subtype 1 (18%) containsedTDP-43-positive and p62-negative inclusion bodies. Subtype 2 (67%) included aggregates that were positive for TDP-43 as well as for p62. Subtype 3 (15%) covered scarce cortical TDP-43 and p62 inclusions. No differences regarding the age of onset, death, or disease duration were observed between the three subtypes, while differences in topographical spread, overall cognitive performances, and co-pathologies were observed, suggesting the involvement of divergent pathomechanisms [9]. Braak et al. proposed that neuronal pathology progresses in a similar spatial sequence but at different rates in distinct ALS patients [68]. Less frequently, ALS pathology without TDP-43-positive inclusions occurs; these cases are mostly linked to variants in the *SOD1* and *FUS* genes and exhibit inclusions of SOD1 (1%) and FUS (2%) [10,13], respectively. Comparing six cases of *FUS* ALS, Mackenzie et al. described two distinct pathological patterns [12]. Cluster 1 contained patients carrying the *FUS* mutation p.P525L with an early onset and was characterized by basophilic and FUS-positive neuronal cytoplasmic inclusions. Cluster 2 included patients carrying the *FUS* mutation p.R521C and later disease onset. These patients were neuropathologically characterized by tangle-like FUS-positive cytoplasmic inclusions in neurons, but also in glial cells. Moreover, wild-type SOD1 is observed in inclusion bodies of motor neurons of sALS patients without known genetic variation, as well as in patients with mutations other than *SOD1*, e.g., *C9orf72* [69,70]. In addition to the aggregated protein, the distribution of protein inclusions across the cortex–spinal-cord axis differs between genetic forms. While in *FUS* and *SOD1* ALS, less cortical protein aggregation occurs, ALS with TDP-43 proteinopathy rather presents a widespread inclusion pattern across the cortex–spinal-cord axis. Interestingly, the extent of TDP-43 accumulation shows great interindividual variation, including among patients of the same genotype [11]. Some postmortem analyses reveal an inclusion pattern restricted to the spinal cord or the motor cortex [71,72]. Neuroanatomically, the clinical phenotype correlates with the localization of the TDP-43 inclusions in hippocampal cells of sALS cases, which feature dementia alongside the motor neuron syndrome. Those patients with dendrospinal pTDP-43 pathology in particular were characterized by a poor prognosis [73]. As an example of the other concomitant neuropathology in ALS, the study of Coan and Mitchell identified neurofilament tangles (78%), as well as amyloid-beta (35%), tau (17%), alpha synuclein (0.04%), and Lewy body formation (11%) inclusion in a population of 46 ALS patients. They even described that 20% of sALS patients form an Alzheimer Disease (AD) subgroup. This subgroup clinically but also pathologically met the criteria for AD, with the presence of β-amyloid (80%) and a dominant pattern of neurofibrillary tangles (100%—especially in the amygdala and hippocampus), and presented with a later age of disease onset [14]. The study also highlighted that compared to limb onset, bulbar onset correlated more strongly with anterior and lateral corticospinal tract degeneration and later age at onset. Postmortem tissues of patients with a rapid disease progression showed a more extensive TDP-43-proteinopathy in the motor cortex [74]. However, it remains somewhat unclear whether divergent inclusion patterns are always the result of different rates of disease progression or whether they are the result of heterogeneous spreading on a spatial level. Taken together, the pathological heterogeneity in ALS potentially reflects the involvement of distinct underlying molecular pathological mechanisms that could play a determining role in the development of treatment strategies. Despite these apparent neuropathological differences, it is important to mention that from a clinical point of view, at an advanced stage of the disease the different subtypes of the disease become indistinguishable. Patients all present with tetraparesis and hypoventilation as well as advanced dysphagia and dysarthria. However, before this advanced stage, different clinical phenotypes can be described.

## 4. Clinical Heterogeneity

### 4.1. Phenotypical Heterogeneity

Typically, the average age at diagnosis of ALS is 50–60 years, with tendency to develop sooner in fALS cases (47–52 years old) compared to sALS cases (58–63 years old) [75]. Usually, a distinction is made to discriminate late-onset ALS (after age 40) from early-onset ALS (before age 40, ~10%). Another ~1% is classified as juvenile-onset ALS, as these patients develop first symptoms before the age of 25 (Figure 1c(i)) [76]. While some mutations have been clearly identified as being the cause of juvenile-, early-, or late-onset ALS, it is mostly unclear what factors influence the age of onset [6]. The diagnosis of ALS is supported by signs of upper motor neuron (UMN) and lower motor neuron (LMN) involvement present in neurological assessment or electrodiagnostic testing in the absence of other clinical abnormalities [77]. The most often-used diagnostic criteria for ALS are the revised El Escorial criteria, which were established in 2000. The criteria subcluster probabilities of the diagnosis dependent on the involvement of different regions (bulbar, cervical, thoracic, and lumbosacral) [78]. Though they are widely used in the inclusion criteria of clinical trials, sensitivity to the diagnosis, especially in early stages of the disease where clinical heterogeneity is very prominent, is limited. To improve sensitivity, the Awaji criteria, electromyographic criteria, and subsequently the simplified Gold Coast diagnostic criteria were developed [79]. For the assessment of severity and progression, staging systems have been developed. To date, either the diagnostic delay or the average decline in the revised ALS functional rating scale (ALS-FRS-R) have been used to describe progression rate. Great variability exists in the rate of progression (Figure 1c(ii)). Poor prognosis and fast progression are associated with patients whose ALS diagnosis has been given less than 8 months after symptom onset, or among patients losing more than 1.4 points/month in the ALS-FRS-R [80]. Bulbar site of onset as well as older age at disease onset have also been associated with fast rates in decline in the ALS-FRS-R [81]. Specific staging systems, one of the most used being King’s clinical staging, can provide information on the stage of the disease [82]. According to the involvement of different anatomical regions, stages correlating well with disease progression and prognosis are defined. 

Depending on the involvement of distinct regions, different phenotypes of ALS are recognized. The spinal form is characterized by spinal onset and involvement of UMN and LMN; the bulbar phenotype is characterized by bulbar onset and involvement of UMN and LMN; the flail arm and the flail leg phenotype is characterized by progressive, predominantly proximal weakness, wasting in the upper or lower limbs, and slower progression of disease to other regions, as well as by progressive muscular atrophy (PMA), where involvement of mainly the LMN is present, and primary lateral sclerosis (PLS), where involvement of mainly the UMN is present. The latter two are very distinct phenotypes of usually slower progression than classical ALS, and are usually accepted to be distinct clusters on both ends of the spectrum of UMN and LMN involvement in ALS (Figure 1c(iii) Motor) [15,16].

Different clinical spreading patterns can be associated with different anatomical spreading patterns, and interestingly, also survival rates. Maranzano et al. demonstrated in a large Italian cohort that in patients with UMN impairment, mostly vertical (e.g., from upper to lower limb) and non-contiguous (e.g., distal foot paresis to distal arm paresis) spreading occurs. Horizontal disease spreading (e.g., left limb to right limb), however, was correlated with LMN involvement. Importantly, vertical spreading patterns were associated with reduced survival. The authors suggest these observations contribute to the hypothesis that one pathomechanism involved in the genesis of ALS involves local diffusion of toxic factors [83]. Gromicho et al. described similar patterns in their cohort of over 1300 ALS patients: progression was more likely to occur in close spinal regions. Early bulbar involvement was less likely if the primary region of onset was in a limb. Contrary to the findings of Maranzano et al. [83] this spreading pattern was found to be independent of the primary involvement of UMN and LMN [84].

While the staging systems aim to stratify disease progression mainly by the progressive involvement of additional anatomical regions, other studies have aimed to subcluster patients according to other non-anatomical features. Westeneng et al. developed the ENCALS score, a personalized prediction model based on clinical parameters, such as age of onset, diagnostic delay, progression rate, forced vital capacity, bulbar onset, definite ALS according to the El Escorial criteria, the presence of frontotemporal dementia, or a *C9orf72* repeat expansion. Five clusters were defined according to the predicted progression rate based on the factors at the date of diagnosis [85]. Faghri et al. in another approach used machine learning to define subclusters of patients, depending on selected clinical features. Out of 42 features fed to the algorithm, 11 features were incorporated: the anatomical level at onset, the site of symptom onset, the site of onset, the weight at diagnosis, the El Escorial category at diagnosis, the ALS-FRS-R part 1 score for speech, the time from symptom onset to first ALS-FRS-R measurement, the smoking status, the age at symptom onset, the rate of body-mass index decline per month, and the forced vital capacity percentage at diagnosis. The model identified subclusters of patients which closely resembled the clinical cluster of the Chiò classification system (bulbar, classical (spinal), flail arm, flail leg, pyramidal, respiratory), emphasizing the importance of the phenotypic subclusters [17].

Cognitive impairment associated with frontotemporal dementia (FTD) is observed in up to 50% of the patients with ALS, and up to 30% of the patients with frontotemporal dementia develop motor dysfunction [86]. It is widely accepted that both diseases exist on a spectrum where the individual patient may have variable features from both diseases (Figure 1c(iii) Behavioral/Cognitive). ALS with cognitive or behavioral changes that do not fulfill formal diagnostic criteria for FTD can be grouped into one of these three categories: (1) ALS with behavioral impairment (ALS-bi), (2) ALS with cognitive impairment (ALS-ci), and (3) ALS with cognitive and executive dysfunction [19]. Lulé et al., however, demonstrated a distinction between the evolution and features of the cognitive profile of patients with motor-predominant ALS and the behavioral variant of FTD. Firstly, a different temporal evolution of symptoms was noted: consistent with the name, in ALS, behavioral symptoms developed after motor symptoms, whereas in FTD, they developed in advance of them. Secondly, while in motor-predominant ALS, mostly minor symptoms (e.g., apathy) developed, ALS-FTD patients also showed plus symptoms (e.g., disinhibition), which more closely resembled the cognitive profile of FTD patients, arguing for the presence of two distinct cognitive phenotypes in the ALS and FTD groups [18]. Ahmed et al. showed that ALS-FTD patients showed different median survival rates depending on the onset of symptoms. Cognitive-onset ALS-FTD was associated with a longer median survival than motor-onset ALS-FTD but shorter median survival than behavioral variant FTD, suggesting possible linkage to physiologic and motor changes of the pathology in these cases [87]. Recent work by Carbayo and colleagues showed a high rate of 35.5% of ALS patients exhibiting neuropathologic features of FTD. Motor symptoms in the bulbar region were more prominent in those patients, although no differences in survival were observed. Because of the highly heterogeneous groups reflecting the MND–FTD disease spectrum, no subclusters could be further identified. Interestingly, a subgroup of six patients was identified who showed pathology restricted to the hypoglossal nucleus and/or anterior horns of the medulla with a matching LMN predominant phenotype [88]. 

Some of the symptoms can be captured by digital biomarkers and be used for tracking disease progression. Yunusova et al. used software-based speech and pause analyses of a simple reading task to discriminate between ALS patients with mild, respiratory, bulbar (with oral-motor deficit), or respiratory signs. They were able to distinguish patients with bulbar ALS from respiratory ALS [89]. Milella et al. used acoustic voice parameters to distinguish ALS from control as well as patients with predominantly upper or lower motoneuron involvement [90]. Accelerometer data assessing physical activity has been shown to correlate well with the ALS-FRS-R and disease progression in various studies [91,92]. In another study, Kelly and colleagues aimed to develop a digital platform to capture multiple symptoms, such as physical activity, heart rate variability, and speech. All assessed data were correlated to gold standard measurements of ALS progression. Multiple parameters of daily activity as well as speech endpoints correlated with sub-scores as well as the total ALS-FRS-R [93]. Digital outcome measures are increasingly used in clinical studies and could potentially reduce sample size due to decreased variability in the data collected [94].

Taken together, there is emerging evidence of different phenotypic subclusters in ALS depending on site of onset, clinical progression, and associated non-motor symptoms, such as FTD-like cognitive deficits. Evidence shows distinct progression and survival probabilities in the different clusters, making the anatomical and biochemical understanding of the processes behind these distinct phenotypes important for patient communication regarding prognosis and the course of the disease, as well as for the development of potentially targeted and individualized treatment options. There also is a yet unmet need for more objective quantitative markers that might hint at the underlying disease pathobiology and predict the efficacy of therapeutic approaches [95]. 

### 4.2. Heterogeneity in Electrophysiologic Findings

In electrophysiological studies, resting-state electroencephalography (EEG) displays a characteristic pattern of increased connectivity between frontocentral/parietal regions and between bilateral regions over motor areas in ALS patients [96,97]. Correlation with structural magnetic resonance imaging (MRI) from the same patient shows that disease-specific structural degeneration in motor areas and corticospinal tracts parallels a decrease in neural activity over scalp motor areas, while the EEG over the scalp regions associated with less extensively involved extra-motor regions on MRI exhibit significantly increased neural communication. Dukic et al. recently identified four subclusters based on patterns of disruption in brain networks, which did not recapitulate common clinical phenotypes or disease burden, but were associated with the rate of progression and survival [20]. Cluster 1 showed an increase in beta-band spectral power in the frontotemporal network, whereas clusters 3 and 4 showed decreased power in the same network, but decrease in gamma-band synchrony in the frontotemporal network for cluster 3, and increase in gamma-band co-modulation in the frontoparietal network for cluster 4. Similarly, cluster 2 showed an increase in alpha-band synchrony in the somatomotor network. Applying functional scores, the found clusters were characterized clinically: cluster 1 was characterized by an intact language domain and cluster 2 by the preservation of executive domain. Cluster 4 was primarily characterized by impairments in bulbar function, verbal fluency, executive functions, and memory, as well as the shortest survival time. Interestingly, *C9orf72* patients did not form one separate cluster, suggesting diverging network impairments caused by the same genetic mutation [20]. To sum up, the identified patterns of network disruption provide additional information as to what can be gathered from clinical evaluation alone, and possibly reflect premorbid patterns of network function and integrity.

Motor unit number estimation (MUNE) methods, especially the promising MUNE technique motor unit number index (MUNIX), are valuable tools for diagnosis and assessment of disease progression in ALS. Derived from the area or amplitude ratio of maximal compound muscle action potential (CMAP) to the average single motor unit potential, MUNE evaluates the number of viable motor units in a studied muscle, which decreases over time in ALS [98]. A motor unit comprises the anterior horn motor neuron, the motor nerve, the neuromuscular junction, and all muscle fibers innervated by the one motor neuron [99]. While MUNE is a more sensitive marker of motor neuron loss than (maximal) CMAP amplitude alone, a CMAP scan as a detailed stimulus response curve visualizes individual motor unit potentials and quantifies motor unit loss and reinnervation [95,100,101]. MUNE is considered a biomarker for the involvement of LMN and shows no significant correlations with cortical atrophy in MRI studies or other signs of UMN involvement. Baumann et al. found that the half-life of motor units and the accumulation of large motor units varies among subclusters of ALS. The increase in motor unit action potential (MUAP) size in the early stages of lesion, reflecting re-innervation, was much greater in LMN predominant subclusters than typical ALS, suggesting that a slower disease progression allows collateral sprouting to occur in these patients [21]. This shows that heterogeneity in ALS in terms of lower motor neuron involvement and reinnervation might contribute to diverging progression rates and survival and could hint at underlying pathophysiological differences.

### 4.3. Heterogeneity in PNS and CNS Imaging

The peripheral nervous system (PNS) can be morphologically evaluated using quantitative ultrasound or magnetic resonance neurography (MRN). While MRN studies inconsistently demonstrate T2 hyperintensities in cervical nerve roots and upper and lower extremity nerves [102,103], ultrasound studies concordantly report cross-sectional area (CSA) reduction in ALS, which is suggestive of PNS degeneration [22,104]. However, not all ultrasound studies confirm peripheral nerve atrophy, and individual patients can display normal or even enlarged nerve CSA (reviewed in [22]). Such cases might be indicative of an inflammatory ALS subcluster. Schreiber et al. therefore classified ALS patients into suspected PNS degeneration vs. PNS inflammation constellations on the basis of their upper limb nerve CSA and albumin ratio (Figure 1c(iv) PNS). They combined the results of nerve ultrasound with the patients’ cerebrospinal fluid (CSF) albumin/serum albumin ratio and found a constellation of peripheral nerve inflammation in up to 21% of ALS patients [23]. Interestingly, all *SOD1* mutation carriers in the study were found in the PNS inflammatory subcluster. The authors stated that inflammation and degeneration might rather be interpreted as a continuum representing stages of the disease.

Imaging of the central nervous system (CNS) using brain MRI revealed cortical thinning of motor and frontotemporal regions and loss of white matter integrity of connections linked to the motor cortex in ALS. Hereby, cortical atrophy is highly variable in terms of degree and distribution (Figure 1c(iv) CNS). Taken together, the extent of cortical atrophy correlates to the extent to which different body regions are affected in ALS [24,25,26]. Patients with a *C9orf72* hexanucleotide repeat expansion frequently exhibit symptoms from the frontotemporal dementia spectrum and are characterized by earlier age of onset and shorter survival and thus a more aggressive form of ALS. The morphological correlate using brain MRI is widespread gray and white matter involvement (including the thalamus as gatekeeper of large-scale cortical networks) at early disease stages and an extensive loss of white matter integrity over time [27,28]. Accordingly, ALS patients with cognitive and behavioral symptoms exhibit widespread cerebral changes extending beyond the primary motor areas, including frontotemporal regions [25]. Van der Burgh et al. also found that patients with spinal onset displayed widespread white matter involvement at baseline and gray matter atrophy over time, whereas patients with bulbar onset started out with prominent gray matter involvement. They hypothesized a dying-backward process of UMNs in spinal-onset patients [25].

Glucose metabolism in different brain areas can be monitored through the Fluorodeoxyglucose *F18 (18F-FDG)* radiotracer with the positron emission tomography (PET). Hypometabolism is observed in frontal, motor, and occipital cortex and hypermetabolism in midbrain, temporal pole, and hippocampus in patients with ALS compared to controls. Comparing spinal- and bulbar-onset ALS, a similar metabolic pattern was found, but bulbar-onset patients had a relative hypometabolism in frontal, prefrontal, and parietal regions, more rostral than spinal-onset patients [29,30], suggesting a differential metabolic state (Figure 1c(iv) CNS). Again, *C9orf72* ALS patients show a more widespread hypometabolism, including the anterior and posterior cingulate cortex, insula, caudate, thalamus, and the left frontal and superior temporal cortex [31]. This could reflect either a more aggressive disease course or an alternative underlying pathogenic mechanism.

Taken together, cerebral-imaging abnormalities in terms of atrophy and hypometabolism correspond to clinical phenotypes regarding the involvement of body regions, cognitive deficits, and genetics.

### 4.4. Metabolic Heterogeneity

Metabolic perturbations usually occur early during the course of ALS and might be a predictor of disease outcome [105]. Weight loss of both fat mass and fat-free mass is common in ALS. This mass loss is believed to be the result of a hypermetabolic state. Hypermetabolism is defined by a significant increase in measured resting energy expenditure (REE) relative to predicted resting energy expenditure. Hypermetabolism is also present in the normal population, but it is observed in a significantly higher proportion in the ALS population [106]. Increase in REE as well as decline in body mass index (BMI) in ALS is associated with a more aggressive disease and a shorter survival [107,108].

Surprisingly, a review summarizing the different metabolic alterations identified in ALS demonstrated that most of the data collected were either insufficient or non-reproducible [109]. It is tempting to speculate that ALS heterogeneity might be the main reason for this non-reproducibility. This idea is strengthened by recent studies clearly identifying metabolic subclusters, firstly among ALS patients but also among presymptomatic gene carriers (Figure 1c(v)) [32,33,34]. In 2022, Cattaneo et al. demonstrated that ALS patients can be divided into three main metabolomic profiles, according to REE calculated via indirect calorimetry [32]. From a cohort of 847 ALS patients, they indeed identified hypermetabolism in 40%, a number that is consistent with similar studies [106]. However, they further identified that 10% of the cohort surprisingly demonstrated a hypometabolic profile, while the remaining 50% demonstrated a normal metabolic profile. Hypometabolic ALS patients had a later onset of the need for gastrostomy, noninvasive ventilation, and tracheostomy placement, and significantly outlived both patients with normal metabolic profiles and those with hypermetabolism. Other studies confirmed a greater involvement of LMN, which causes muscle atrophy and weight loss, in the hypermetabolic ALS subcluster [106]. Interestingly, in the study by Cattaneo et al., *C9orf72* mutation carriers were almost exclusively restricted to the normal or hypermetabolic profile, thus suggesting potential implications of genetic mutations in ALS metabolic alteration. Further studies focusing on presymptomatic gene carriers indeed demonstrated that *C9orf72* and *SOD1* mutation carriers had inverted metabolic profiles [34]. While *C9orf72* carriers showed lower BMI, lower fasting serum glucose, and higher HDL compared to controls, *SOD1* carriers demonstrated higher BMI, higher fasting serum glucose, and lower HDL compared to healthy controls. The study even suggested that metabolic heterogeneity might further exist among carriers of mutations in the same gene, as carriers of benign, slowly progressive *SOD1* mutations demonstrated significant metabolic changes (in HDL and LDL) compared to those with other *SOD1* mutations. Interestingly, in this study, the REE seems to be lower in both *C9orf72* and *SOD1* mutation carriers compared to controls, in contrast to what is observed in the ALS symptomatic phase [33]. As the expected disease onset approaches, REE tends to increase, thus suggesting that the REE measure might be a biomarker of phenoconversion. When studying heterogeneity in ALS, the metabolic profile is crucial for addressing the nutritional and metabolic needs of individuals and has implications for the development of supportive therapies. Because preclinical studies in the G86R and G93A *SOD1* mouse model showed that a calorie-dense high-fat diet leads to weight gain and a delay in disease progression, while calorie restriction reduces survival, clinical studies in humans quickly followed [110,111,112]. Indeed, treatment with high-caloric and/or high-fat nutritional interventions demonstrate benefit in survival and reduced loss of body weight, mainly in patients with a fast-progressing ALS [113,114]. As described before, fast progressors are mainly associated with a hypermetabolic profile that indeed should benefit strongly from energetic supply [32].

## 5. Molecular Heterogeneity

### 5.1. Fluid-Based Biomarkers

Being one of the most promising fluid-based biomarkers in ALS, neurofilaments (Nf) as markers of axonal damage have been studied extensively. In general, Nf levels are increased in ALS compared to healthy controls and similar diseases, and their prognostic value nowadays is indisputable [115,116]. Nf levels are highest in ALS patients with a more aggressive disease course, which correlates to certain clinical and/or genetic phenotypes (i.e. bulbar-onset, UMN-predominant and classic ALS, presence of *C9orf72* hexanucleotide repeat expansion). This has been reviewed in detail elsewhere [117,118].

As neurofilaments are non-specific markers of axonal damage, they are not suitable for the differentiation of subclusters using underlying pathobiological or molecular mechanisms or genetics. The same applies to other well-known ALS markers, such as markers of muscle metabolism (creatine kinase, creatinine, uric acid), and nutritional status (protein, albumin etc.) measured in routine blood workup, even if they correlate with motor neuron loss, disability, and survival [119,120,121,122,123]. Concentrations of the extracellular domain of the common neurotrophin receptor (p75^ECD^) in urine have been reported to reflect motor neuron loss and disease progression as they increase over the course of disease. This is in contrast to neurofilaments, which remain mostly stable, and thus p75^ECD^ is a promising pharmacodynamical biomarker that could be applied to monitor the effect of experimental therapeutics [124,125]. However, there are no studies on the ability of p75^ECD^ to differentiate subgroups of ALS with different responses to distinct treatment approaches. 

A systemic immune response with elevated calcitonine-related peptide (CRP), among other indicators, has been described before in patients with ALS and was correlated with the degree of disability [126]. As patients who exhibit elevated serum CRP concentrations progress more rapidly, the existence of an inflammatory subcluster has been proposed that is potentially responsive to immunomodulatory treatment [35]. In line with this, the transcription factor FoxP3, which is required for the immunosuppressive function of regulatory T-lymphocytes, is reduced in rapidly progressing ALS [36]. Further, CHIT1 as a marker of microglial/macrophage activation is increased in the CSF of ALS patients and correlates with disease progression [37]. Fast progressors had significantly higher levels of CHIT1 than slow progressors, and in presymptomatic gene-mutation carriers, CHIT1 levels rose before onset of symptoms [127,128]. Differences in cytokine levels in ALS also correlate negatively with survival [129]. All could therefore be used as markers for immune activation—and possibly identify an “inflammatory ALS cluster”—which could be relevant for patient stratification in therapeutic trials targeting immunologic mechanisms.

Recently, there has been an increasing use of plasma and CSF extracellular vesicles (EVs) in the identification of novel biomarkers. EVs can be released from different cell types (e.g., neurons, glial cells) in the brain and secreted into the periphery. Some ALS-associated proteins (e.g., SOD1, FUS, TDP43, and C9orf72 expansion DPRs) and previously described biomarkers (e.g., phospho-Nf, CHIT1) can be identified from these EVs. This has been extensively reviewed elsewhere [130]. Interestingly, Pasetto et al. discovered that plasma EV size distribution associated with its levels of cyclophilin A (PPIA) enabled the distinction between fast and slow disease progressors [131].

Thus, there is a variety of fluid-based biomarkers for ALS that can aid diagnosis and predict progression, whose comprehensive description is outside of the scope of this review [132]. So far, studies using these fluid-based biomarkers to identify ALS subclusters and show different treatment responses are lacking.

### 5.2. Omics Based Biomarkers

#### 5.2.1. Transcriptomic

Recent studies analyzed the RNA expression profiles in the postmortem brain tissue of ALS patients to provide better insights into molecular clusters within the disease (Figure 1d). Aronica et al. and Morello et al. analyzed the whole-genome expression profile of 31 sALS patients’ postmortem motor cortices and succeeded in highlighting two molecularly different sALS subclusters using unsupervised hierarchical clustering [38,133]. While cluster 1 was mainly associated with increased immune response pathways, energy metabolism, and oxidative phosphorylation, cluster 2 demonstrated decreased activation of these pathways. Moreover, cluster 2 also demonstrated specific activation of genes involved in apoptosis, cell cycle, axonal transport pathways, and decreased expression of cytoskeleton-related genes. Dysregulation of genes associated with cell adhesion pathways and extracellular matrix were also specific to cluster 2. Thus, these studies highlighted two ALS subclusters with a somewhat inverse molecular pattern. Another study from the same group demonstrated a specific genomic signature between the two ALS clusters when analyzing copy number of variants [134]. Based on the same unsupervised clustering, Cagnata et al. combined the transcriptomic data from the 31 ALS motor cortices of the previous studies with the transcriptomic data of 30 ALS spinal cords [135]. By analyzing the complete spectrum of genes encoding splice factors (396 genes), the study revealed contrasting regulation between the clusters, with cluster 2 being more affected, and with greater separation between the clusters when the expression of genes in the motor cortex was analyzed, while in the spinal cord, a less pronounced distinction between the two clusters was seen.

The work of Gomes et al. regrouped the transcriptomic profiles from the prefrontal cortexes (PFC) of 51 patients with ALS as well as the PFCs of the four most common ALS mouse models [39]. Hierarchical clustering of the human samples highlighted the presence of four different major subclusters with specific alterations (C1–C4). They found that the regulation of the immune response distinguished ALS patients best (C1 and C2 vs. C3 and C4), whereas second-level arborization was mainly driven by extracellular matrix components (C1 vs. C2) and synaptic function and protein folding (C3 vs. C4). Interestingly, this study pinpointed that the gene profiles of each ALS cluster correlated most with one of the four mouse models. Thus, mouse models cannot fully recapitulate ALS, but might rather represent the alteration of specific ALS-associated pathways that are impaired in some but not all ALS subclusters. These findings argue in favor of a judicious choice of animal models in preclinical studies of ALS, as well as the use of drugs in human subgroups presenting alterations similar to the animal model where the drug has been validated. Another important point of the study of Gomes et al. is the observed sex-specific difference, not only on a transcriptional level, but also on the proteomic and microRNAome level. ALS sex differences have also been reported in other studies [136,137,138]. Since the prevalence of ALS is slightly higher in males than females (~60% vs. 40%), and this proportion is usually respected in clinical trials, results might be somewhat driven by male ALS patients. Thus, we might gain more knowledge regarding the implication of sex in treatment response if clinical studies reported results stratified by sex.

Tam et al. and Eshima et al. characterized the transcriptome of the frontal and motor cortices from 77 and 208 (including the 77 patients from Tam et al. [9]) unique ALS patients, respectively [40,139]. Using unsupervised hierarchical clustering, they identified three main distinct ALS molecular subclusters. While the ALS-Glia group was characterized by massive glial activation (ALS-Glia), a second subcluster presented with prominent activation of oxidative stress and altered synaptic signaling (ALS-Ox) and a third subcluster presented mainly transcriptional dysregulation (ALS-TD). ALS-Ox, ALS-TD, and ALS-Glia subclusters were identified with stable ratios of 3:1.4:1 in Tam et al. and 3:1.9:1 in Eshima et al. [139] Interestingly, the ALS-Glia subcluster was associated with a significantly decreased survival. Using the same hierarchical clustering protocol, Marriott et al. also divided 112 postmortem motor cortices in three subclusters, similarly to the two previous studies [140]. A specific cell type involvement was attributed to each cluster: clusters 1 (neuronal signaling), 2 (excitotoxicity), and 3 (inflammation) demonstrated higher contributions of the neuronal, endothelial, and glial cells, respectively, thus indicating different main modes of pathogenesis in each ALS cluster. By including clinical and DNA methylation measures to this data, the authors also identified further cluster differences. Compared to cluster 1, cluster 2 demonstrated a better clinical and omics-based outcome (e.g., higher age at death, lower biological and transcriptional aging calculation, higher mitochondrial DNA copy numbers), while cluster 3 was found in between. The authors suggest the better outcome of patients in cluster 2 might be related to a history of Riluzol usage with a better treatment effect, since glutamatergic excitotoxicity is one of the main pathophysiological mechanisms affected in this cluster. Interestingly, almost all *C9orf72* ALS patients were found in cluster 1 (18/23). This study also reproduced the clustering method in other independent ALS cohorts, first on 93 other postmortem motor cortices, and second on the peripheral blood mononuclear cells (PBMC) of 412 ALS patients.

Other studies also attempted to identify ALS clusters, but from blood samples. In 2019, Swindell et al. aimed to analyze the gene expression of the whole blood of 396 ALS patients [41]. Using the microarray technique, they highlighted two main ALS subclusters with different immune cell signatures. While one group demonstrated high expression of genes expressed from myeloid derived cells (neutrophils, monocytes, dendritic cells, macrophages, platelets, red blood cells, and eosinophils), the second subcluster showed enrichment for genes expressed from lymphoid-derived cells (CD4+ and CD8+ T-cells, gamma delta T-cells, B-cells, and NK-cells). Moreover, the authors described a trend for a “high inflammation” profile in the myeloid patient group.

In a more recent study, RNA sequencing of whole-blood samples of 96 sALS patients revealed four distinct molecular subclusters [42]. In this clustering approach, immune response also played an important role since it was oppositely regulated in cluster 0 and cluster 3. Conversely, clusters 1 and 2 mainly showed opposing dysregulation for genes associated with catabolic processes, metabolism, RNA splicing, cellular transport, and DNA damage.

All studies analyzed gene expression profiles in ALS. However, differences across studies can in part be attributed to the technique used (e.g., microarray vs. RNA-seq) or by the type of postmortem tissue (frontal cortex, motor cortex, or spinal cord) or biofluid (whole blood or PBMC) used. While all studies on postmortem tissues focused on sALS, the group of Eshima et al. also included ALS-FTD and ALS-AD samples in their cohort. Furthermore, whereas Aronica et al. focused on the motor cortex and Gomes et al. on the PFC of ALS patients, Eshima et al. collected and integrated transcriptomic data from both the frontal and motor cortexes, thus further complicating the possibility for direct comparisons between the different studies [39,139].

#### 5.2.2. Proteomic

While transcriptomic studies provide a comprehensive overview of gene expression, they cannot predict associated protein expression. In contrast, proteomics gives a lower identification rate (<10% of total proteins) but provides direct information on possible drug targets. Thus, proteomics is a good choice for biomarker research [141]. One study from Xu et al. demonstrated that the plasma proteomic profile of ALS correlates with some clinical features [43]. Indeed, they identified 20 proteins significantly dysregulated in ALS patients with or without cognitive impairment. in the latter case, these proteins were mainly involved within the coagulation (downregulated) and immune pathways (upregulated). The same study also identified a single protein, the complement C1s subcomponent, to be significantly dysregulated between *C9orf72* expansion carriers and non-carriers. In another study from Vu et al., the authors identified a list of proteins differentially expressed between fast and slow progressors. Interestingly, proteins associated with fast progressors were part of the immune response pathway, while pathways related to synaptogenesis and glycolysis/gluconeogenesis were downregulated [44]. These two studies again highlight the importance of monitoring the immune response pathway for ALS clustering. Interestingly, while transcriptomic clustering did not succeed in explaining patient clinical profile, proteomic clustering demonstrated some correlation with some clinical features (concomitant cognitive impairment, progression rate) [43,44]. Thus, studying the proteomic profile of ALS patients might give access to more pertinent information. While there is limited literature available regarding ALS proteomic heterogeneity and clustering, it poses an important future topic, as it could have a direct impact on new drug development.

#### 5.2.3. Lipidomic

Studying lipidomics in ALS can also provide better understanding of ALS clinical heterogeneity. From two different CSF lipidomic prediction models, Blasco et al. were able to predict the evolution of the ALS-FRS-r score, the force vital capacity (FVC), the variation of the BMI, and the duration of survival (based on median survival) [45]. A more recent study from Sol et al. studied the lipid composition of plasma and CSF of ALS patients. They first observed that plasma contained more differential lipids than CSF and that few differential similitudes were observed between the two biofluids [46]. Using hierarchical clustering of 25 plasma lipid species, they were able to identify three clusters associated with the main involvement at onset (spinal, bulbar, and respiratory) and with 25 other plasma lipids were able to differentiate between fast and normal progressors. These data were partially reproduced in CSF with different lipids. Interestingly, both lipidomic studies identified that lower levels of plasma triglycerides were associated with a better prognosis [45,46].

In conclusion, deciphering the molecular landscape of ALS patients could yield valuable information regarding what defines clinical clusters (e.g., identification of differential molecular expression depending on the site of onset or progression), and could also help to define new molecular clusters according to recurrent pathway alterations observed within patients. 

## 6. Summary and Conclusions

As research progresses, the complexity and heterogeneity of ALS becomes more evident. In this review, we have highlighted this for different aspects of the disease. 

Subclustering of ALS based on genetics can be made with a steadily increasing number of known ALS-related genes. Genetics are becoming increasingly important in therapy development, since various mutations are associated with specific pathomechanisms. The genotype correlates to some extent with clinical and epidemiological data as well as with the metabolic status and some biomarkers, thus forming a specific subgroup. The various mutations, however, are closely linked to the neuropathological subclusters, whereby the distribution of inclusions across the cortex–spinal-cord axis appears to go beyond genetics. Initial symptoms and anatomical spreading of affected regions is associated with different rates of disease progression and disease severity. The degree of involvement of UMN and LMN is a further factor contributing to disease progression and phenotype. Machine learning algorithms using a wide variety of clinical and epidemiological parameters have been developed to classify patients. The subclusters closely resemble already-established phenotypical classification systems. Brain MRI and FDG-PET show high variance in the degree and distribution of cortical atrophy and glucose metabolism, corresponding to clinical phenotypes and genetic background. In contrast, EEG can identify distinct patterns of network disruption that do not recapitulate clinical phenotypes. MUNE can distinguish subclusters of ALS with different degrees of reinnervation, and ultrasound of the PNS might be able to detect an inflammatory subtype. ALS patients can be subclustered according to their metabolic profile (hypermetabolic versus hypometabolic versus normal metabolic profile), with hypermetabolism being associated with greater involvement of LMN, weight loss, faster disease progression, and shorter survival time. Metabolic profiles seem to identify ALS subgroups independently from genetic or clinical subgroups and provide promising therapeutic approaches. Regarding molecular classification, while the postmortem cortex gives better insight into the pathology in the most vulnerable region of ALS patients, analyses of blood samples give premortem access to information that is partly derived from the brain. Interestingly, a lot of studies agree on the importance of monitoring genes associated with immune response to differentiate between ALS subclusters. Other pathways, such as genes associated with oxidative stress and phosphorylation, neuronal functions (synaptic signaling/axon transport), and metabolic pathways, were found enriched in more than one study. Further studies on molecular subclusters will be needed for a consensus on how to classify ALS cohorts. 

In light of the growing understanding of heterogeneity in ALS, studies have identified subclusters of patients (for a summary see Table 1). Whether these subclusters represent different aspects of the same disease or diseases of different origin with a similar phenotype remains to be understood. While the different genetic subtypes result in distinct histopathological phenotypes, little is known about why different mutations in the same gene can lead to vastly different disease phenotypes. Imaging and fluid-based biomarkers have started to identify subgroups with the prospect of discovering the causes of different progression rates and severities, although identification of a correlation to distinct clinical and histopathological phenotypes is still lacking. During the last decades, the development of new therapeutic approaches for ALS has been unsuccessful. Recently, efforts have been made to optimize the selection of clinical trial participants, using scores based mostly on clinical parameters [142]. This reflects a growing trend in attempts to divide study populations into subgroups that might be more likely to benefit from a specific treatment; this division potentially reduces the effects of variability in the disease course of the study population. As an example, follow-up studies of Riluzole treatment identified better survival benefits in the ALS subpopulation with bulbar onset [143,144]. While the development of targeted treatments for specific genetic mutations, such as the silencing ASO Tofersen in patients carrying certain pathogenic *SOD1* variants, addresses only a small subset of the ALS population, these individualized treatment approaches show promising initial results. Beyond ASO therapies, some clinical trials have already shown different treatment effects according to the genetic status of patients, as outlined in Section 2.3. Therefore, ALS patients might benefit from systematic genetic analyses and stratification in order to have access to such specific treatment options. Most recently, intermediate and fast progressors were found to profit from treatment with rasagiline, showing prolonged survival and a reduced decline in the ALS-FRS-R after 18 months. In contrast to rasagiline’s efficacy in Parkinson disease, this was independent of existing variances in the *MAOB* and *DRD1* genes [145]. All studies focusing on molecular classification of ALS as well as some imaging and fluid biomarker studies identified a cluster with a high “inflammatory” profile. It is therefore tempting to speculate that drugs targeting the immune system, such as Triumeq (NCT05193994), Masitinib (NCT03127267), MN-166/Ibudilast (NCT04057898), or Verdiperstat (NCT04436510), currently in phase III or II/III clinical trials, might be more successful in the “inflammatory” ALS subclusters, but might have no or even negative impact on other ALS subclusters. In an attempt to pre-select ALS patients with an inflammatory profile, in the Tocilizumab (NCT02469896) and NP001 (NCT02794857) clinical trials, respectively, PBMC gene expression profiling and blood CRP levels were analyzed [146,147]. One of the two studies identified a group of patients categorized as “responders” [147]. Further studies, such as the clinical trial on low dose interleukin 2 (NCT02059759) and the AMX0035 trial, also tried to identify the molecular profiles of these so called “responders” [148,149]. Of course, the identification of responders would be specific to every single treatment, while cluster identification might be suitable for a more generalized usage. While the clinical trial with Edaravone was negative, it would be interesting to know whether the subgroup with a positive response to Edaravone treatment (NCT01492686) [150,151] correlates with the molecular subtype ALS-Ox from the Tam et al. study, and whether this in turn is consistent with the positive clinical effect of Edaravone in fast-progressing ALS patients

While cluster identification based on postmortem tissue is only possible postmortem, and blood mainly gives access to peripheral information, we would greatly benefit from further studies based on CSF samples. ALS subclustering from tear fluid might also be of great promise, as biomarkers from other neurodegenerative diseases have already been identified in this way [152,153]. Moreover, cluster analyses of brain-derived EVs (e.g., proteins, miRNA, lipids, and metabolites) would offer potential insights into the pathological states of the brains of ALS patients via either CSF collection or simple blood or tear fluid collection. All these new discoveries mark the beginning of molecular-based patient classification in clinical trials and indicate that further studies are necessary to better identify suitable ALS clusters and responders.

Currently, we only have limited treatment options available for a very heterogenous disease population (Figure 2a). In this model we observe a dichotomy of responders and non-responders among patients. While some of the non-responders simply demonstrate no benefits from the treatment, others experience adverse reactions, which should be avoided. In this review we mainly described how ALS patients could be classified on the genetic, clinical, or molecular levels. However, a combination of these subclusters might be useful in the development of treatment strategies for the wider ALS population as well as for the individual assessment of the therapeutic benefit. In the future, treatment options should consider subclustering patients in more homogenous groups and allow for a more personalized medicine (Figure 2b). First, we suggest that more systematic genetic testing would allow stratification according to genetic subtype. While *SOD1* mutation carriers can already benefit from Tofersen, other patients with a specific causal mutation will hopefully soon benefit from similar treatment approaches. Second, if no genetic mutation is identified or if no treatment option exists for the identified mutation, patients will then be screened for clinical or molecular features. Most clinical studies already incorporate clinical data into their results, and some of them identified a better effect in bulbar onset (e.g., Riluzole) or fast progressors (e.g., Rasagiline), as already discussed. Molecular groups could be selected by evaluating the expression of a small panel of biomarkers (e.g., 5–10 features–proteins, metabolite, lipids, and/or miRNA via targeted immunoassay or qPCR) that would be linked to the most important altered pathways of each molecular cluster (e.g., high vs. low immune response activation). This molecular data could be analyzed from CSF samples or from blood, or even from other easily accessible biofluids, such as tear fluid. Depending on the results, patients would be assigned to a specific molecular cluster and would receive appropriate treatment. In addition, disease progression should be monitored via longitudinal clinical examination (e.g., affected region, ALSFRS-R) and biomarker quantification (apparative and fluid based) to adjust for treatment at any time. Altogether, we herein suggest three different ways of clustering ALS patients and offering them personalized medicine. We are aware that ALS is a complex disease, and that more subclusters and treatment groups might evolve in the future and that some patients might even benefit from combined treatment strategies. 

In conclusion, the recent advances in the identification of genetic, histopathological, clinical, and molecular heterogeneity in ALS have represented progress towards a more granular understanding of the disease. This opens up the possibility of adapting clinical trial design to fit pre-selected participants based on their ALS subcluster, and therefore offering a personalized treatment approach in an effort to help patients on an individual level. Further work is needed to understand and link the findings on the different aspects of heterogeneity.

## Figures and Tables

**Figure 1 cells-13-00452-f001:**
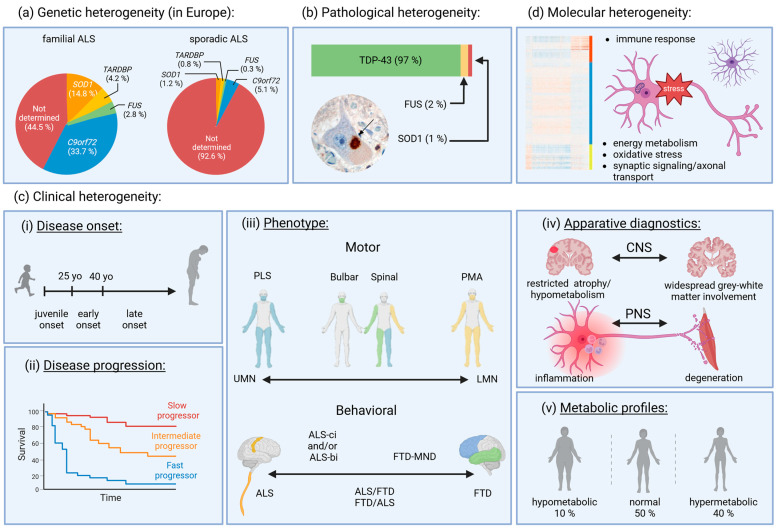
Overview of the different aspects of heterogeneity in ALS. (**a**) Genetic heterogeneity; (**b**) pathological heterogeneity; (**c**) clinical heterogeneity regarding; (i) disease onset; (ii) disease progression; (iii) motor and behavioral/cognitive phenotype; (iv) apparative diagnostics and (v) metabolic profiles; and (**d**) molecular heterogeneity. ALS, amyotrophic lateral sclerosis; *SOD1* and SOD1, superoxide dismutase 1; *TARDBP*, TAR-DNA-binding protein; *FUS* and FUS, fused in sarcoma; *C9orf72*, chromosome open reading frame 72; TDP-43, TAR DNA-binding protein 43; PLS, primary lateral sclerosis; PMA, progressive muscular atrophy; UMN, upper motor neuron; LMN, lower motor neuron; FTD, frontotemporal dementia; MND, motor neuron disease; ci, cognitive impairment; bi, behavioral impairment; bci, behavioral and cognitive impairment; CNS, central nervous system; PNS, peripheral nervous system. Created by Laura Tzeplaeff and Camilla Wohnrade with “BioRender.com”.

**Figure 2 cells-13-00452-f002:**
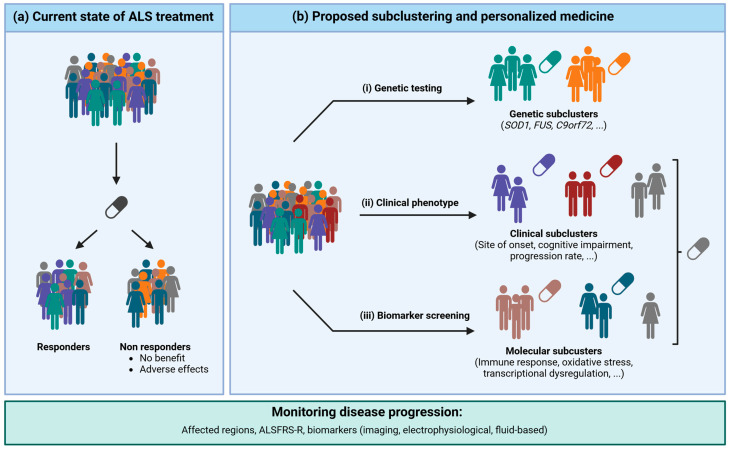
Current state of ALS treatment and proposed subclustering for personalized medicine.

**Table 1 cells-13-00452-t001:** Overview over the different subclusters identified in literature.

Feature	Subclusters	Literature
Genetic	fALS (C9orf72, SOD1, FUS, TARDBP, others)	[5,6]
sALS
Pathology	p62-positive DPRs	[7]
TDP-43 inclusions (TDP-43+ p62-; TDP-43+ p62+; pTDP-43+)	[8,9,10,11]
FUS—with either basophilic or tangle-like inclusions	[12]
SOD1	[13]
ALS with concomitant neuropathology (neurofilament tangles, amyloid-beta, tau, alpha synuclein, and Lewy body formation)	[14]
Clinical		
*Phenotype*	Motor:	[15,16,17]
Spinal
Bulbar
Flail arm
Flail leg
Pyramidal
Respiratory
PMA
PLS
Behavioral and cognition:	[18]
ALS with behavioral impairment
ALS with cognitive and executive dysfunction
ALS with cognitive but non-executive dysfunction

ALS with FTD	[19]
FTD with ALS
*Electrophysiology*	EEG:	[20]
Cluster 1: high beta-band spectral power in the frontotemporal network and intact language domain
Cluster 2: high alpha-band synchrony in the somatomotor network and preservation of executive domain
Cluster 3: low gamma-band synchrony in the frontotemporal network
Cluster 4: high gamma-band comodulation in the frontoparietal network and impairments in bulbar function, verbal fluency, executive functions, and memory, as well as the shortest survival time
EMG:	[21]
UMN-dominant type—less reinnervation
LMN-dominant type—slower disease progression and greater re-innervation
*CNS/PNS imaging*	Ultrasound	[22,23]
Nerve CSA reduction—PNS degeneration
Enlarged nerve CSA—PNS inflammation (SOD1)
Brain MRI	[24,25,26,27,28]
Different extent of cortical atrophy—ALS with cognitive and behavioral symptoms exhibit widespread cerebral changes (C9orf72)
Different extent of white matter involvement

Spinal onset—widespread white matter involvement at baseline	[25]
Bulbar onset—prominent gray matter involvement
Brain 18F-FDG PET	[29,30,31]
Spinal—hypometabolism in frontal, motor, and occipital cortex and hypermetabolism in midbrain, temporal pole, and hippocampus
Bulbar—supplemental hypometabolism in frontal, prefrontal and parietal regions, likely more rostral than spinal
C9orf72—widespread hypometabolism including anterior and posterior cingulate cortex, insula, caudate and thalamus, the left frontal and superior temporal cortex
*Metabolic*	Hypermetabolic	[32]
Hypometabolic
Normal metabolic profile

Presymptomatic C9orf72 metabolic profile	[32,33,34]
Presymptomatic SOD1 metabolic profile (allele specific)
Molecular		
*Fluid biomarkers*	Inflammatory subtype	[35,36,37]
*Omics*	Transcriptomics	
	[38]

Postmortem brain:	
Cluster 1: immune response pathways, energy metabolism and oxidative phosphorylation up	
Cluster 2: apoptosis, cell cycle, axonal transport pathways up and expression of the cytoskeleton down	[39]

C1–C2 vs. C3–C4: active immune response pathway in C3–C4	
C1 vs. C2: extracellular matrix up in C1	
C3 vs. C4: synaptic function and protein folding up in C4	[40]


ALS-Ox: oxidative pathway up	
ALS-TD: transcriptional dysregulation	
ALS-Glia: glial activation	

Blood:	[41]
Myeloide dominant expression pattern	
Lymphoide dominant expression pattern	

Cluster 0 vs. 3: upregulation of the immune response in Cluster 3	
Cluster 1 vs. 2: proteolysis, RNA splicing and cellular transport up in cluster 2 and DNA damage, catabolic processes, and metabolism up in cluster 1	[42]

Proteomic:	

Plasma:	[43]
ALS without or with cognitive impairment: immune pathway up and coagulation pathway down with cognitive impairment	
C9orf72 carriers vs. other ALS: complement C1s subcomponent up in *C9orf72* carriers	
CSF:	[44]
Fast progressor vs. slow progressor: upregulation of immune response pathway and downregulation of synaptogenesis and glycolysis/gluconeogenesis pathway in fast progressors	
Lipidomic	

CSF:	[45]
ALSFRS-R score evolution	
FVC evolution	
BMI variation	
Duration of survival (based on median survival)	

Plasma and CSF:	[46]
Main involvement at onset: panel of 25 most regulated lipids	
Fast vs. normal progressor: panel of 25 most regulated lipids

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
