# Peer review of "Unraveling the Heterogeneity of ALS—A Call to Redefine Patient Stratification for Better Outcomes in Clinical Trials"

_cells, 2024, doi:10.3390/cells13050452_

Round 1
Reviewer 1 Report
Comments and Suggestions for Authors
Authors review very well about “the heterogeneity of ALS”, and I think that this manuscript has some value to be published in Cells.
Comments to the authors.
Minor comments:  
>Reconsider the “Keywords”.
Is it necessary both ALS and amyotrophic lateral sclerosis in key words?
It would be better to add “Clinical trials” in Keywords.
>Figure 1; C) i) > iii)?
>L97; While fALS cases with a FUS mutation > cases of
>The part of L603 to L630 in “Conclusion” might be transferred to “2.3. Treatment options for genetic clusters”.
Author Response
Authors review very well about “the heterogeneity of ALS”, and I think that this manuscript has some value to be published in Cells.
Comments to the authors.
Minor comments:  
1.) Reconsider the “Keywords”. Is it necessary both ALS and amyotrophic lateral sclerosis in key words? It would be better to add “Clinical trials” in Keywords.
The keyword “clinical trials” has been added while “ALS” was removed.
2.) Figure 1;C) i)> iii)?
We thank the reviewer for noticing this mistake. This has been corrected.
3.) L97; While fALS cases with a FUS mutation > cases of
We thank the reviewer for the suggestion. We asked two different native English speakers and they confirmed that “cases with FUS mutation” is also correct. If the reviewer does not mind, we would prefer to keep the sentence as it is.
4.) The part of L603 to L630 in “Conclusion” might be transferred to “2.3. Treatment options for genetic clusters”.
We thank the reviewer for this suggestion. While we have moved the mentions of the UNC13A polymorphism trial to 2.3 we have kept the other studies in the paragraph as they are not exclusively addressing genetic subgroups but subclusters in general and serve as example of clinical trials which already incorporate sub-clustering in their study design. The paragraph has been adapted as follows:
Lines 742-746:
“Beyond ASO therapies, some clinical trials already showed different treatment effects according to the genetic status of patients, as] outlined in section 2.3. “Treatment options for genetic clusters”. Therefore, ALS patients might benefit from systematic genetic analyses and stratification to have access to such specific treatment options.”

Reviewer 2 Report
Comments and Suggestions for Authors
This is a very nicely organized and written review covering the very important topic of heterogeneity of ALS. Figure 1 is very nicely done, summarizing all the different factors contributing to heterogeneity in a concise way. Table 1 is also very useful in tracking down references.
For the sake of completeness, I would like to see some information added on the following topics:
1) The use of digital biomarkers (voice, accelerometers, other wearables, etc.) in tracking the disease progression and the different clusters that exist within each one. This can go under Clinical Heterogeneity after phenotypical heterogeneity.
2) Some information on the increasing use of plasma and CSF exosomes in the identification of novel biomarkers and their potential application in patient stratification.
3) The inclusion of CMAP in the electrophysiology section and some discussion related to clustering.
Finally, it's important to state, at the end of the pathological heterogeneity section, that despite the heterogeneity in the pathology of ALS in early and intermediate stages, at end-stage ALS cases are effectively clinically indistinguishable.
Author Response
This is a very nicely organized and written review covering the very important topic of heterogeneity of ALS. Figure 1 is very nicely done, summarizing all the different factors contributing to heterogeneity in a concise way. Table 1 is also very useful in tracking down references.
For the sake of completeness, I would like to see some information added on the following topics:
1.) The use of digital biomarkers (voice, accelerometers, other wearables, etc.) in tracking the disease progression and the different clusters that exist within each one. This can go under Clinical Heterogeneity after phenotypical heterogeneity.
We have added a paragraph on digital biomarkers in the section on clinical heterogeneity:
Lines 325-339:
“Some of the symptoms can be captured by digital biomarkers and be used for tracking disease progression. Yunusova et al. used software-based speech and pause analyses of a simple reading task to discriminate ALS patients with mild, respiratory, bulbar (with oral-motor deficit) or respiratory signs. They were able to distinguish pa-tients with bulbar ALS from respiratory ALS [62]. Milella et al. used acoustic voice pa-rameters to distinguish ALS from control as well as patients with predominantly upper or lower motoneuron involvement [63]. Accelerometer data assessing physical activity has been shown to correlate well with the ALS-FRS-R and disease progression in vari-ous studies [64,65]. In another study, Kelly and colleagues aimed to develop a digital platform to capture multiple symptoms such as physical activity, heart rate variability and speech. All assessed data was correlated to gold standard measurements of ALS progression. Multiple parameters of daily activity as well as speech endpoints correlat-ed with sub-scores as well as the total ALS-FRS-R [66]. Digital outcome measures are increasingly used in clinical studies and could potentially reduce sample size due to decreased variability in the data collected [67].”
2.) Some information on the increasing use of plasma and CSF exosomes in the identification of novel biomarkers and their potential application in patient stratification.
To our knowledge, studies on ALS clusters only exist on post-mortem brain tissue or whole blood. We did not find any publication aiming to cluster ALS patients either with whole CSF samples or with plasma or CSF exosomes or extracellular vesicles. We agree with the reviewer that this could indeed offer a potential application in ALS patient stratification. Especially brain derived extracellular vesicles can provide information on the pathological state of the brain at a premortem state. We included the two next paragraphs in the revised manuscript:
Lines 533-540:
“Recently, there is an increasing use of plasma and CSF extracellular vesicles (EVs) in the identification of novel biomarkers. EVs can be released from different cell types (e.g.: neurons, glial cells) in the brain and secreted into the periphery. Some ALS associated proteins (e.g.: SOD1, FUS, TDP43 and C9orf72 expansions DPRs) and previously described biomarkers (e.g.: phospho-Nf, CHIT1) can be identified from these EVs. This has been extensively reviewed in elsewhere. Interestingly, Pasetto et al. discovered that plasma EVs size distribution associated to its levels of cyclophilin A (PPIA) allows to distinguish fast and slow disease progressors.“
Lines 769-776:
“While cluster identification based on post-mortem tissue is only possible post-mortem and blood mainly gives access to peripheral information, we would greatly benefit from further studies based on CSF samples. ALS subclustering from tear fluid might also be of great promise, as biomarkers from other neurodegenerative diseases have already been identified in there. Moreover, cluster analyses of brain-derived EVs (e.g.: proteins, miRNA, lipids and metabolites) would offer potential insights into the pathological state of the brain of ALS patients via either CSF collection, or a simple blood or tear fluid collection.“
3.) The inclusion of CMAP in the electrophysiology section and some discussion related to clustering.
We included CMAP in the revised version of our manuscript as follows and shortly discussed the involvement of lower motor neuron degeneration as clustering approach:
4.) Finally, it's important to state, at the end of the pathological heterogeneity section, that despite the heterogeneity in the pathology of ALS in early and intermediate stages, at end-stage ALS cases are effectively clinically indistinguishable.
We thank the reviewer for this important comment. The advanced stage of ALS indeed is clinically very homogeneous despite the heterogeneity in pathology. We included a paragraph in the revised manuscript:
Lines 218-223:
“Despite these apparent neuropathological differences, it is important to mention, that from a clinical point of view, at an advanced stage of the disease the different subtypes of the disease become indistinguishable. Patients all present with tetraparesis and hypoventilation as well as advanced dysphagia and dysarthria. However, before this advanced stage different clinical phenotypes can be described.”

Reviewer 3 Report
Comments and Suggestions for Authors
This is in general a comprehensive review covering the different factors underlying the current efforts to stratify ALS patients. Section 4.1 about different phenotypical rating systems used clinically is especially well-written. However, there are still room to improve for sections 5, as lots of biochemical, proteomic and lipidomic biomarkers were not discussed.
Page 3, the second example may need some more explanation to avoid misleading the readers. The main pathomechanism underlying SOD1 ALS patient is believed to be the gain-of-function toxicity of the mutated SOD1 protein, while the loss of the antioxidation function of SOD1 may have a minor role. This is because (i) in humans, a lack of correlation was found between SOD1 dismutase activity and aggressiveness of clinical phenotypes; (ii) in mice, a lack of overt ALS-like phenotype was found in SOD1 null (SOD1−/−) animals; (iii) transgenic mouse models over-expressing mutant human SOD1 have increased SOD1 activity and a loss of motor neurons that models human ALS (DOI: 10.1093/brain/awt097). It is a fact that oxidative stress level increased in brain tissue of SOD1 ALS patients, yet it could be attributable to other factors like increased amount of myeloperoxidase mentioned by the authors, rather than the compromised dismutase activity of SOD1.
Page 3, for the third example, please discuss more about the physiological role of STMN2 protein in neurite outgrowth and that mutant STMN2 expression is associated with retraction of motor neurons from innervated neuromuscular junctions in Drosophila (DOI: 10.1523/JNEUROSCI.2024-11.2011).
Page 9-10, section 5.1 missed lots of biofluid biomarkers reported previously. For example, Urinary p75ECD levels have been reported to be a potential diagnostic biomarker and a progression indicator for ALS patients (DOIs: 10.1080/21678421.2021.1990345; 10.1212/WNL.000000000000374). 4-HNE (a reactive aldehyde generated by peroxidation of cell membrane lipids) levels have also been reported to increase in cerebrospinal fluid, blood of ALS patients. Please refer to this review for more examples (DOI: 10.3389/fneur.2019.00291).
Page 10, section 5.2: This section only focused on transcriptomics-based approach to stratify ALS patients, while proteomics and lipidomics approaches were not discussed. I suggest that the authors either change the subheading to “transcriptomics-based biomarkers” or expand this section to discuss studies adopting other omics approaches as well. Please refer to these literatures for detail (DOIs: 10.3390/proteomes11010001; 10.1038/s41598-017-17389-9; 10.1093/braincomms/fcab143).
Minor issues:
Page 3, line 95, 104, 111: Please add “the” before “first”, “second”, “last”.
Page 3, line 106-108: Please rephrase this sentence to “Especially in brain samples from SOD1 ALS patients, with >90% of neurons CEP positive and ~5-fold higher levels of CEP deposition compared to those from C9orf72 ALS and ALS without a known mutation.”
Page 3, line 118: Please change the heading to “Treatment options targeting different genetic subtypes”.
Page 4, line 120: Please change “clustering” to “Treating”.
Page 4, line 129: Please change “acts as a promoter of” to “promotes”. The reader may confuse it with the promoter which drives gene expression.
Page 7, line 290-291: This sentence is somewhat disconnected from the above context. Maybe the authors can elaborate more.
Page 7, line 319-320: Please change “viable motor neurons” to “viable motor units”. Please also explain that “Motor unit is a functional entity consisting of the anterior horn motor neuron, motor nerve, neuromuscular junction and all muscle fibers that are innervated by one motor neuron” (DOI: 10.1007/s13311-016-0473-z).
Page 8, line 358: A parenthesis is missing.
Page 8, line 365: Please change “Glucose metabolism shows a clear pattern with” to “Glucose metabolism in different brain areas can be monitored through”.
Page 10, line 472: Please change “the cytoskeleton” to “cytoskeleton-related genes”.
Page 10, line 474: Please change “inverted” to “inverse”.
Page 11, line 512-513: Please change “the shortest disease duration” to “the fastest disease progression”.
Author Response
This is in general a comprehensive review covering the different factors underlying the current efforts to stratify ALS patients. Section 4.1 about different phenotypical rating systems used clinically is especially well-written. However, there are still room to improve for sections 5, as lots of biochemical, proteomic and lipidomic biomarkers were not discussed.
Page 3, the second example may need some more explanation to avoid misleading the readers. The main pathomechanism underlying SOD1 ALS patient is believed to be the gain-of-function toxicity of the mutated SOD1 protein, while the loss of the antioxidation function of SOD1 may have a minor role. This is because (i) in humans, a lack of correlation was found between SOD1 dismutase activity and aggressiveness of clinical phenotypes; (ii) in mice, a lack of overt ALS-like phenotype was found in SOD1 null (SOD1−/−) animals; (iii) transgenic mouse models over-expressing mutant human SOD1 have increased SOD1 activity and a loss of motor neurons that models human ALS (DOI: 10.1093/brain/awt097). It is a fact that oxidative stress level increased in brain tissue of SOD1 ALS patients, yet it could be attributable to other factors like increased amount of myeloperoxidase mentioned by the authors, rather than the compromised dismutase activity of SOD1.
We thank the reviewer for this important comment. We totally agree with that the toxic gain of function of SOD1 in SOD1-associated ALS is the main pathomechanism. Although oxidative stress is a major mechanism in ALS in general, the CEP level varies between ALS subgroups. To clarify that this is not due to the loss of SOD1 function in SOD1 ALS, we have added this information in the paragraph as follows:
Lines 110-118:
“It is a fact that oxidative stress level is increased in brain tissue of SOD1 ALS patients, but this could also be attributable to other factors than SOD1 activity itself. Indeed, SOD1 ALS patients particularly exhibit high CEP levels in brain samples with >90 % CEP positive neurons and ~5-fold higher levels of CEP deposition compared to those from C9orf72 ALS and ALS without a known mutation. This was attributed to the amount of myeloperoxidase (MPO), as substantially more myeloperoxidase expressing cells were detected in SOD1 (~8-fold) and C9orf72 (~2-fold) ALS patients’ brains com-pared to controls [16]. It must be emphasized that the SOD1 activity is independent of this phenomenon as it does not correlate clinically with the aggressiveness of ALS [17].”
Page 3, for the third example, please discuss more about the physiological role of STMN2 protein in neurite outgrowth and that mutant STMN2 expression is associated with retraction of motor neurons from innervated neuromuscular junctions in Drosophila (DOI: 10.1523/JNEUROSCI.2024-11.2011).
Indeed, the STMN2 gene is an interesting topic in ALS. Especially in TDP-43 proteinopathy where missplicing of STMN2 could be observed leading to low Stathmin-2 levels and thus to a decline of axonal transport, stability and growth in motor neurons. Therefore, we modified the following paragraph and due to the high value of treatment strategies added the ongoing Stathmin-2 ASO trial in the section 2.3. “Treatment options for genetic clusters”.
Lines 121-125:
“Stathmin-2 protein is abundant in motor neurons and plays an important role in microtubule dynamics, axonal transport, stability and regeneration as well as for neuro-muscular junction stability [20]. The reported casecontrol study supports the hypothesis that allele length is a determinant of disease risk and that stratification into risk genotype groups might be a useful tool for cohort selection.”
Lines 141-149:
“Another ASO therapy concerns Stathmin-2 (STMN2) (NCT05633459), which acts as an important splice target of TDP-43 (TAR DNA-binding protein 43) and thus the loss of nuclear TDP-43 as observed in ALS leading to a Stathmin-2 miss-splicing. While the clinical trial of this ASO therapy is ongoing, the Stathmin-2 levels have been restored in preclinical studies and could transform the therapeutical options for ALS patients.”
Page 9-10, section 5.1 missed lots of biofluid biomarkers reported previously. For example, Urinary p75ECD levels have been reported to be a potential diagnostic biomarker and a progression indicator for ALS patients (DOIs: 10.1080/21678421.2021.1990345; 10.1212/WNL.000000000000374). 4-HNE (a reactive aldehyde generated by peroxidation of cell membrane lipids) levels have also been reported to increase in cerebrospinal fluid, blood of ALS patients. Please refer to this review for more examples (DOI: 10.3389/fneur.2019.00291).
We thank the reviewer for this important notification. We are aware that and we didn’t aim to include all (potential) biomarkers for ALS in this review, but focused on the fluid biomarkers that are useful for subclustering ALS patients. As p75ECD concentrations seems to be a promising pharmocodynamic biomarker, which might be useful to monitor the effect of novel treatments, we included a section as follows:
Lines 510-517:
“Concentrations of the extracellular domain of the common neurotrophin receptor (p75ECD) in urine have been reported to reflect motor neuron loss and disease progres-sion as they increase over the course of disease. This is in contrast to neurofilaments, which remain mostly stable, and makes p75ECD a promising pharmacodynamical bi-omarker, which could be applied to monitor the effect of experimental therapeutics [97,98]. However, there are no studies on the ability of p75ECD to differentiate sub-groups of ALS with different responses to distinct treatment approaches.”
We further clarified that we did not aim to comprehensively describe all ALS fluid-based biomarkers, since we focus on biomarkers that could help for ALS clustering:
Lines 541-544:
“Thus, there is a variety of fluid-based biomarkers for ALS, which can aid diagnosis and predict progression , whose comprehensive description is out of the scope of this review [108]. So far, studies using these fluid-based biomarkers to identify ALS subclusters and showing different treatment responses are lacking.“
Page 10, section 5.2: This section only focused on transcriptomics-based approach to stratify ALS patients, while proteomics and lipidomics approaches were not discussed. I suggest that the authors either change the subheading to “transcriptomics-based biomarkers” or expand this section to discuss studies adopting other omics approaches as well. Please refer to these literatures for detail (DOIs: 10.3390/proteomes11010001; 10.1038/s41598-017-17389-9; 10.1093/braincomms/fcab143).
We greatly thank the reviewer for identifying these important missing points and for suggesting some important literature. We now realized that the part 5.2 initially focused on modeling molecular clusters based on transcriptomic data. However, proteomic and lipidomic data could also give important information regarding clinical clustering and this information was missing from our initial manuscript. To correct for this, we suggest two new subsubsection “proteomic” and “lipidomic” and table 1 has been updated accordingly with the suggested literature and some more.
Lines 640-685:
“5.2.2 Proteomic
While transcriptomic studies provide a comprehensive overview of gene expres-sion, they cannot predict associated protein expression. In contrast, proteomics gives a lower identification rate (<10% of total proteins) but provides direct information on possible drug targets. Thus, it makes proteomics a good choice for biomarker research [122]. One study from Xu et al. demonstrated that the plasma proteomic profile of ALS correlates with some clinical features [123]. Indeed, they identified 20 proteins signifi-cantly dysregulated between ALS patients with or without cognitive impairment. The latter were mainly proteins involved within the coagulation (downregulated) and im-mune pathways (upregulated). The same study also identified a single protein, the complement C1s subcomponent, to be significantly dysregulated between C9orf72 ex-pansion carriers and non carriers. In another study from Vu et al, the authors nicely identified a list of proteins differentially expressed between fast and slow progressors. Interestingly, proteins associated with fast progressors were part of the immune re-sponse pathway, while pathways related to synaptogenesis and glycoly-sis/gluconeogenesis were downregulated [124]. These two studies again highlight the importance of monitoring the immune response pathway for ALS clustering. Interest-ingly, while transcriptomic clustering did not succeed to explain patient clinical pro-file, proteomic clustering demonstrated some correlation with some clinical features (concomitant cognitive impairment, progression rate) [123,124]. Thus, studying prote-omic profile of ALS patients might give access to more pertinent information. While there is limited literature available regarding ALS proteomic heterogeneity and clus-tering it poses an important future topic as this can have a direct impact for new drug development.
5.2.2 Lipidomic
Studying lipidomic in ALS can also provide better understanding of ALS clinical heterogeneity. From two different CSF lipidomic prediction models Blasco et al. were able to predict the evolution of the ALS-FRS-r score, the force vital capacity (FVC), the variation of the BMI and the duration of survival (based on median survival) [125]. A more recent study from Sol et al, studied the lipid composition of plasma and CSF of ALS patients. They first observed that plasma contained more differential lipids than CSF and that only little differential similitudes were observed between the two bioflu-ids [126]. Using hierarchical clustering of 25 plasma lipid species they were able to identify 3 clusters associated to the main involvement at onset (spinal, bulbar and res-piratory) and with 25 other plasma lipids differentiate between fast and normal pro-gressors. These data were partially reproduced in CSF with different lipids. Interesting-ly, both lipidomic studies identified that lower levels of plasma triglycerides was asso-ciated with a better prognosis [125,126].
In conclusion, deciphering the molecular landscape of ALS patients might give precious information regarding what defined clinical clusters (e.g.: identification of differential molecular expression depending on the site of onset or progression), but al-so help to define new molecular clusters according to recurrent pathway alterations observed withing patients.”
Minor issues:
Page 3, line 95, 104, 111: Please add “the” before “first”, “second”, “last”.
This has been updated.
Page 3, line 106-108: Please rephrase this sentence to “Especially in brain samples from SOD1 ALS patients, with >90% of neurons CEP positive and ~5-fold higher levels of CEP deposition compared to those from C9orf72 ALS and ALS without a known mutation.”
This has been rephrased.
Page 3, line 118: Please change the heading to “Treatment options targeting different genetic subtypes”.
The heading has been changed.
Page 4, line 120: Please change “clustering” to “Treating”.
This has been changed.
Page 4, line 129: Please change “acts as a promoter of” to “promotes”. The reader may confuse it with the promoter which drives gene expression.
We agree that this sentence was confusing. We changed it accordingly.
Page 7, line 290-291: This sentence is somewhat disconnected from the above context. Maybe the authors can elaborate more.
We agree with the reviewer. This sentence referred to the next paragraph. After assessing the train of thought in the paragraph we do not think, this sentence is needed to connect to the next paragraph. It has been removed.
Page 7, line 319-320: Please change “viable motor neurons” to “viable motor units”. Please also explain that “Motor unit is a functional entity consisting of the anterior horn motor neuron, motor nerve, neuromuscular junction and all muscle fibers that are innervated by one motor neuron” (DOI: 10.1007/s13311-016-0473-z).
We changed „viable motor neurons“ to „viable motor units” and added a sentence to explain motor unit according to the reviewers suggestions:
Lines 380-382:
“A motor unit comprises of the anterior horn motor neuron, the motor nerve, the neu-romuscular junction and all muscle fibres innervated by the one motor neuron.”
Page 8, line 358: A parenthesis is missing.
We added the parenthesis.
Page 8, line 365: Please change “Glucose metabolism shows a clear pattern with” to “Glucose metabolism in different brain areas can be monitored through”.
We thank the reviewer for pointing out this unprecise way of describing the nature of Fluorodeoxyglucose F18 radiotracer positron emission tomography. We changed the sentence accordingly.
Page 10, line 472: Please change “the cytoskeleton” to “cytoskeleton-related genes”.
This has been changed.
Page 10, line 474: Please change “inverted” to “inverse”.
It has been adapted.
Page 11, line 512-513: Please change “the shortest disease duration” to “the fastest disease progression”.
Eshima et al. correlate the identified subtypes with the disease duration, which was significantly shorter in the ALS-glia group. The progression rate, usually as a measure of loss of points in the ALS-FRS-R over the course of time, was not mentioned. We have rephrased the sentence to make it clearer, that the ALS-Glia subgroup had a shorter survival after onset of symptoms.
Lines 597-598:
“Interestingly, the ALS-Glia subcluster was associated with a significantly decreased survival.”

Reviewer 4 Report
Comments and Suggestions for Authors
I have enclosed comments and suggestions,

This paper is an important review of the issues surrounding ALS diagnosis and therapy. It is an important paper to publish. Not to say that these issues have not been discussed previously.
Author Response
This is a very useful manuscript but needs more focus.
1.) Tofersen treatment is mentioned (lines 121-122), however no mention is made about why it may not work in many subjects with the SOD1 genetic form of ALS. This is an important point. In the reviewer’s experience, the results of targeting SOD1 have not been impressive.
We thank the reviewer for this important comment. Not all mutations found in SOD1 are gain of function mutations and other factors may contribute to the disease pathogenesis (Ruffo et al. Genes 2022). Only some variants have been verified in vitro. Differences in treatment response may vary as a result of many other factors as for the course of disease there is large heterogeneity in the group of mutation carriers. The causes are less well understood and need to be further investigated. We included a paragraph in the revised manuscript:
Lines 135-138:
“Differences in treatment response vary within a subgroup of mutation carriers due to large heterogeneities. For example some ALS patients with SOD1 mutations exhibit no toxic gain of function pathogenesis and probably do not benefit from the reduced SOD1 levels in Tofersen treatment.“
2.) Line 128: In ALS…not of ALS
This has been changed.
The authors point out the pathology of ALS can be quite diverse reflecting the heterogeneity of the disease.
3.) 149-153: Three subtypes of TDP-43 pathology are mentioned with and without P-62 showing the heterogeneity of disease pathology. It is a very diverse set of diseases.
We kindly thank the reviewer for this comment. In the study of Tan et al. from 2023 only post-mortem tissues of ALS patients were examined. Patients with cognitive or behavioral impairment that met criteria for ALS-FTD were not included in this study. Within the ALS patients with TDP-43 pathology, the 3 mentioned subtypes could be observed, namely TDP-43 without p62, TDP-43 with p62 and scarce cortical TDP-43 and p62 inclusions. The clinical phenotype apart from the performance in memory tests did not reveal a difference between these three pathological phenotypes.
4.) 158: Forms of ALS without TDP-43 pathology also exist.
Thank you for this important comment. We agree and find this aspect very important to mention, which is why we indeed included in our initial manuscript a paragraph about the histopathology of SOD1- and FUS-associated cases.
Lines 182-192:
“Less frequently, ALS pathology without TDP-43-positive inclusions occurs: these cases are mostly linked to variants in the SOD1 and FUS genes and exhibit respectively inclusions of SOD1 (1 %) or FUS (2 %) […]”
5.) 161-169: Various pathological variants of PUS ALS are also described.
More examples of pathological variants exist along location of pathology which can carry different prognoses.
We thank the reviewer for this important comment. Indeed, depending on the affected region, pathological variants have a different prognosis. Therefore, we added a paragraph in the revised manuscript with a few examples:
Lines 202-213:
“As an example for the other concomitant neuropathology in ALS, the study of Coan and Mitchell identified neurofilament tangles (78 %), as well as amyloid-beta (35 %), tau (17 %), alpha synuclein (0,04 %), and Lewy body formation (11 %) inclusion in a population of 46 ALS patients. They even described that 20 % of sALS patients form an Alzheimer Disease (AD) subgroup. This subgroup meets clinically but also pathologically the criteria for AD, with the presence of β-amyloid (80 %) and a dominant pattern of neurofibrillary tangles (100% - especially in the amygdala and hippo-campus) and present with a later age of disease onset [41]. The study also highlight that compared to limb onset, bulbar onset correlated more strongly with anterior and lateral corticospinal tract degeneration and later age at onset. Post mortem tissues of patients with a rapid disease progression showed a more extensive TDP-43-proteinopathy in the motor cortex [42].”
6.) As outlined in Table 1, there are many other heterogeneities including phenotype, electrophysiology, ultrasound, CNS/PNS imaging, metabolics, molecular, inflammatory biomarkers/omics. The authors have clearly outlined that ALS involves a vast number of underlying pathophysiological mechanisms leading to ALS. As pointed out by the authors, identifying subclusters of ALS will be very important in terms of finding effective therapies. This is in fact what precision medicine is all about. No combination of targeted therapies is going to be successful in arresting the progression of all ALS cases. I think the authors should have a figure or a more involved discussion of how one should and will be approaching treatments for ALS in the future. What tests will be done? Will brain biopsies be necessary to evaluate pathology? What blood tests need to be done? What CSF tests will need to be done to design a therapy? All these questions/procedures/diagnostic tests should be very helpful at designing therapies for the major heterogeneity of the disease.
We agree with the reviewer that a detailed discussion and hypothesis of a potential scenario of future ALS treatment will strengthen our discussion. Therefore, we added a paragraph to the conclusion and summed up the main information in a new Figure 2:
Figure 2: Current state of ALS treatment and proposed subclustering for personalized medicine.
Lines 784-814:
“In the current state, we only have limited treatment options available for a very heterogenous disease population (Figure 2a). In this model we observe a dichotomy of responders and non-responders among patients. While some of the non-responders demonstrate simply no benefit of the treatment, other experience adverse reactions, which should be avoided. In this review we mainly described how ALS patients could be classified on the genetic, clinical or molecular level. However, a combination of these subclusters might be useful in the development of treatment strategies for the wider ALS population as well as for the individual assessment of the therapeutic benefit. In the future, treatment options should consider subclustering patients in more homogenous groups and allow for a more personalized medicine (Figure 2b). First, we suggest that a more systematic genetic testing will allow to stratify according to genetic subtype. While SOD1 mutation carriers can already benefit from Tofersen, other patients with a specific causal mutation will hopefully soon benefit from similar treatment approaches. Second, if no genetic mutation is identified or if no treatment option exists for the identified mutation, patients will then be screened for clinical or molecular features. Most clinical studies already incorporate clinical data to their results and some of them identified a better effect in bulbar onset (e.g.: Riluzole) or fast progressors (e.g.: Rasagiline), which we already discussed. Molecular groups could be selected by evaluating the expression of a small panel of biomarkers (e.g.: 5-10 features - proteins, metabolite, lipids and/or miRNA via targeted immunoassay or qPCR) that would be linked to the most important altered pathways of each molecular cluster (e.g.: High vs. low immune response activation). This molecular data could be analyzed from CSF samples or ultimately from blood or even other easily accessible biofluids such as tear fluid. Depending on the results, patients will be assigned to a specific molecular cluster and will receive appropriate treatment. In addition, disease progression should be monitored via longitudinal clinical examination (e.g.: affected region, ALSFRS-R) and biomarker quantification (apparative and fluid based), to adjust for treatment at any time. Altogether, we herein suggest three different ways of clustering ALS patients and offering them personalized medicine. We are aware that ALS is a complex disease and more subclusters and treatment groups might evolve in the future and that some patients might even benefit from combined treatment strategies.”
I think the authors have done a great job of explaining the issues related to ALS and its therapies, but could expand some more as suggested above to make their case clearer.
I recommend the article for publication with minor revisions.
